# Clinical and Non-Clinical Cardiovascular Disease Associated Pathologies in Parkinson’s Disease

**DOI:** 10.3390/ijms241612601

**Published:** 2023-08-09

**Authors:** Bonn Lee, Charlotte Edling, Shiraz Ahmad, Fiona E. N. LeBeau, Gary Tse, Kamalan Jeevaratnam

**Affiliations:** 1School of Veterinary Medicine, Faculty of Health and Medical Sciences, University of Surrey, VSM Building, Daphne Jackson Road, Guildford GU2 7YW, UK; bonn.lee@surrey.ac.uk (B.L.); c.edling@surrey.ac.uk (C.E.); sa0079@surrey.ac.uk (S.A.); gary.tse@kmms.ac.uk (G.T.); 2Biosciences Institute, Faculty of Medical Sciences, The Medical School, Newcastle University, Framlington Place, Newcastle upon Tyne NE2 4HH, UK; fiona.lebeau@newcastle.ac.uk; 3Kent and Medway Medical School, University of Kent and Canterbury Christ Church University, Canterbury CT2 7FS, UK

**Keywords:** Parkinson’s disease, non-motor syndrome, cardiovascular risk, cardiovascular disease, dysautonomia, autonomic nervous system, alpha-synuclein, electrophysiology

## Abstract

Despite considerable breakthroughs in Parkinson’s disease (PD) research, understanding of non-motor symptoms (NMS) in PD remains limited. The lack of basic level models that can properly recapitulate PD NMS either in vivo or in vitro complicates matters. Even so, recent research advances have identified cardiovascular NMS as being underestimated in PD. Considering that a cardiovascular phenotype reflects sympathetic autonomic dysregulation, cardiovascular symptoms of PD can play a pivotal role in understanding the pathogenesis of PD. In this study, we have reviewed clinical and non-clinical published papers with four key parameters: cardiovascular disease risks, electrocardiograms (ECG), neurocardiac lesions in PD, and fundamental electrophysiological studies that can be linked to the heart. We have highlighted the points and limitations that the reviewed articles have in common. ECG and pathological reports suggested that PD patients may undergo alterations in neurocardiac regulation. The pathological evidence also suggested that the hearts of PD patients were involved in alpha-synucleinopathy. Finally, there is to date little research available that addresses the electrophysiology of in vitro Parkinson’s disease models. For future reference, research that can integrate cardiac electrophysiology and pathological alterations is required.

## 1. Introduction

### 1.1. Research Aim


“Cardiovascular diseases are critical and life-threatening in Parkinson’s disease patients. Even so, the association of cardiovascular disease with PD is not an area that has been thoroughly studied. In this narrative review, we aim to highlight cardiovascular disease as a non-motor feature of Parkinson’s disease. We have reviewed known cardiovascular features and abnormalities developing in Parkinson’s disease patients. Topics listed and discussed include; ECG alterations in PD patients, alpha-synuclein (α-synuclein) involvement in cardiomyopathic and neurocardiac lesions and finally, we have reviewed fundamental research covering electrophysiological studies using in vitro PD models.”


### 1.2. Overview of Parkinson’s Disease

Parkinson’s disease (PD) is the second most common age-related neurodegenerative disorder impacting patients and their families, and causes our society a substantial economic burden [1]. The prevalence of PD is about 3% of the population by the age of 65 and 5% of individuals over 85 years, and is twice as high in men as in women, while mortality is higher in women [2]. Loss of dopaminergic neurons in the substantia nigra pars compacta (SNpc) is found to be the distinctive feature of neurodegeneration in PD, which may involve many other neuronal systems and organs. During the neuronal loss, the formation of abnormal protein aggregates called Lewy bodies has become a pivotal pathological marker for PD [3]. However, the main cause of neuronal degeneration in PD remains unclear.

### 1.3. Aetiology of Parkinson’s Disease

The aetiology of PD is multifactorial. Sporadic and familial Parkinsonism are considered primary PD while Parkinsonism induced by other diseases or toxins are secondary. The diagnosis of PD is based on symptoms, patient history, and genetic factors. Sporadic PD is typically more late-onset than familial PD and is considered the most frequent type of the disease, while familial PD may develop earlier and is associated with the mutation of risk factor genes [4]. When comparing the dominance between these two types of PD, most cases are sporadic, while 10–30% of patients are associated with a family history of PD [5]. However, only a small fraction is determined by an identifiable monogenic genetic cause in familial PD patients; to date, more than 20 genetic mutations have been identified to be associated with PD [6]. Interestingly, symptom-wise, there are no significant differences in clinical phenotype between sporadic and familial PD [7].

### 1.4. Two Categories of Symptoms in Parkinson’s Disease

James Parkinson first described the disease symptoms as “Essay on the Shaking Palsy”. In the earliest descriptions of symptoms, PD can be understood as the classic triad of Parkinsonism: resting tremor, bradykinesia, and rigidity. Therefore, patients have been clinically diagnosed by their motor dysfunctions [8]. The use of the drug levodopa as a dopaminergic therapy has relieved motor symptoms in PD patients successfully; however, after the dramatic benefit from the initial treatment, patients can develop dopa-related side effects such as motor fluctuations and dyskinesias, and dopa-resistant motor or non-motor symptoms [9]. Recent studies have indicated that PD is a constellation of multiple neurological symptoms beyond the classic triad of Parkinsonism [10]. Thus, it is now widely accepted that PD symptoms embrace a wide range of non-motor symptoms (NMS) as well as motor dysfunctions [8]. Case control studies reported that 88 to 100% of patients developed more than one type of NMS [11,12,13].

## 2. Non-Motor Symptoms of Parkinson’s Disease

### 2.1. Major Phenotypes of Non-Motor Symptoms in Parkinson’s Disease

Despite the high prevalence of NMS, the manifestation of the symptoms can vary widely across individuals, leading to different clinical presentations. Hyposmia, impaired vision, hallucinations, pain, anxiety, excessive sweating, depression, cognitive dysfunction, dementia, sleep disturbances, bladder hyperreflexia, and constipation are all considered non-motor symptoms commonly associated with PD [8,14]. These NMS may also occur in 68–88% of the normal population; however, PD patients tend to have a larger number of NMS and the degree of symptoms are more frequent and severe than in the healthy population [12,15]. Even though NMS are generally highly prevalent among PD patients, the symptoms are often not obvious to the patients themselves as a concomitant problem of their PD [16,17]. Nevertheless, the impact on quality of life can be substantial if these symptoms are not properly managed. Cardiovascular, gastrointestinal and urogenital autonomic dysregulation are all predominant in PD [18].

### 2.2. Research Spotlight on the Early Onset of Non-Motor Symptoms

In regard to NMS, researchers have focused on early diagnosed patients because, unlike motor symptoms, NMS can be identified in the prodromal or early phase of the disease [19]. In most cases of PD, the patients are diagnosed with PD when they develop motor symptoms; however, these motor symptoms are generally only recognized when patients have already lost 60–80% of the midbrain dopaminergic neurons [20]. On the contrary, NMS can develop up to 10 years prior to the first diagnosis, indicating that pre-diagnostic neurodegeneration may progress systematically before patients present with motor syndromes [21]. Considering that PD patients are likely to be initially misdiagnosed for their NMS complaints, NMS could instead be a promising early diagnosis biomarker of PD, enabling neuroprotection to be introduced at an earlier stage in Parkinsonism development [14].

## 3. Cardiovascular Disease as Non-Motor Features of Parkinson’s Disease

### 3.1. Cardiovascular Associated Non-Motor Symptoms of Parkinson’s Disease

Cardiovascular symptoms are relatively less frequent in PD compared to other NMS categories [12]. Cardiovascular symptoms have not commonly been associated with NMS in earlier PD studies; however, recent studies propose a link between PD and cardiovascular diseases [22,23,24] (Table 1).

Given that recognizing NMS before motor symptoms appear can give an advantage in the diagnosis of PD at an early stage, clarifying any cardiovascular pathology in PD and its clinical course is essential. For example, Akbilgic et al. (2020) have attempted to develop a prediction model for pre-motor phase PD by using electrocardiograph (ECG) data obtained from patients [25]. Their stepwise regression model uses probabilistic symbolic pattern recognition (PSPR) of transition points on an ECG. However, this prediction model of the prodromal PD cases has not yet been externally validated. Considering that PD-associated cardiovascular risks increase with age, and that cardiovascular disease strongly contributes to the mortality of PD patients, cardiovascular symptoms need to be carefully monitored in PD. Nowadays, stroke, cardiomyopathy, coronary disease, dysrhythmia, and sudden cardiac death are all known to develop in PD patients [22,24].

Below, the pathology of the vascular system and the heart itself will be addressed; dysrhythmia and sudden cardiac death (SCD) will then be discussed.

### 3.2. Increased Cardiovascular Risk in Parkinson’s Disease Patients

There is an emerging consensus that PD patients carry an increased risk of developing cardiovascular disease as the disease progresses [22,26,27,28,29]. Li et al. (2018) reported the prevalence of stroke and coronary artery disease (CAD) in two different population-based cohort groups, revealing that stroke and CAD may be a potential component in the pathogenesis of PD [24]. Liang et al. (2015) reported that PD patients showed a higher risk of acute myocardial infarction [26]. Swallow et al. (2016) investigated 3019 cases with recent-onset PD by applying the calculator-prediction algorithm (QRISK2) for cardiovascular disease, which included variables such as ethnicity, age, sex, and underlying medical conditions. The result demonstrated that 62.5% of patients had medium to high range cardiovascular risk, which is higher than the general population [27]. The prevalence of cardiovascular disease in the patients was associated with increasing age (*p* < 0.001) and higher severity of motor symptoms (*p* < 0.001) [27]. These reports further support the hypothesis that cardiovascular disease is associated with PD.

### 3.3. Cardiovascular Risks Associated with Other Non-Motor Symptoms

Several studies have reported that cardiovascular disease may be associated with and present together with other NMS, and it can also be present as a consequence of other NMS [37,38]. First, the presence of rapid eye movement (REM) sleep disorder was associated with an increased risk of developing stroke in PD, suggesting that the sleep dysfunction may cause various autonomic consequences [37]. Furthermore, cognitive decline in PD patients is known to be associated with orthostatic hypotension (OH), but whether the relationship is causative or associative remains unclear [38]. OH is the most generally recognized cardiovascular-associated disorder in PD, and approximately 60% of PD patients will develop OH [39]. The link between OH and Lewy Body may reflect common pathology affecting the brain and autonomic nervous system (ANS) [38]. Cardiovascular symptoms can be a direct effect of PD; in addition, case-control studies suggest that it can also be a side effect of autonomic dysfunction (dysautonomia)-associated NMS. In this regard, understanding cardiovascular association with PD may help shed light on the systemic autonomic dysregulation of PD in general.

## 4. Cardiovascular Pathologies in Parkinson’s Disease Patients

### 4.1. Cardiac Abnormalities as a Non-Motor Feature in Parkinson’s Disease

Parkinson’s disease patients develop cardiac abnormalities such as cardiomyopathies or heart failure [22,23,28,29]. Gonçalves et al. (2021) reported that cardiac abnormalities are very common, and affect approximately 80% of PD patients [23]. Similarly, the prevalence of heart failure is more than two times higher in PD patients compared to that of the non-PD group [29]. Zesiewicz et al. (2004) therefore already proposed in 2004 that heart failure could be a NMS of PD due to its considerably high prevalence and severity in elderly PD patients [29]. As heart failure is regarded as one of the main causes of death in PD, it is of the utmost importance to recognize cardiovascular disease as NMS [40]. In addition, one clinical study has indicated that PD directly caused structural abnormalities in the patients’ hearts. Flores et al. (2017) reported increased structural abnormalities in 50 PD patients compared to a healthy control group. Increased risks for left ventricular hypertrophy and diastolic dysfunction were also demonstrated by measuring the ventricular mass index and the left atrial volume in PD patients [28]. Moreover, cardiomyopathy has been episodically observed in case reports from patients with Parkinsonism [22,41]. In addition, independent of cardiomyopathy, abnormalities of heart rhythm have also been reported in a case-controlled study; an early phase-PD group exhibited significant comorbidity with atrial fibrillation, while late phase-patients did not [42]. Cardiomyopathies may lead to changes in the electrophysiological properties of the heart, as reviewed below. Although the aetiology of heart failure in PD is unclear, some research groups have suggested autonomic dysregulation, while others focused on channelopathies caused by PD pathology [22,23,29]. Cardiac denervation or channelopathies could both interfere with the regulation of blood pressure or cardiac rhythm, which could be a possible link to heart failure in PD, but further studies are required.

### 4.2. Cardiac Associated Consequences of Parkinson’s Disease

Since the recognition of cardiac-associated consequences such as electrophysiological alterations following PD development, several studies have aimed to further elucidate cardiac impairments caused by PD [22,43,44]. There are various parameters available to screen cardiac health in PD patients, including heartbeat analysis, electrocardiograph (ECG), echocardiograph, blood pressure, and cardiac magnetic resonance imaging (MRI) scintigraphy [22]. Among these, heart rate variability (HRV) and ECGs are considered simple, yet powerful diagnostic tests to evaluate heart function and cardiac autonomic dysregulation [43,44]. Alterations of HRV or ECG are now widely documented in PD patients. Additionally, cardiac sudden unexpected death in Parkinson’s disease (SUDPAR) is discussed below because of its association with both heart and autonomic dysregulation.

### 4.3. Electrocardiographic Consequences

Recent studies have focused on ECG of PD patients, with the hope of validating a tool for early diagnosis of PD. A range of ECG parameters associated with PD are illustrated in Figure 1. Here, we have categorized a range of previous studies covering electrocardiogram in PD patients by the described ECG parameters. RR interval and heart rate variability, PR interval, and QT interval have all been assessed with a focus on alterations in ECG patterns and the relationship between ECG and the severity of disease (Table 2).

*RR interval* and *heart rate variability* (*HRV*) indicate the intrinsic activity of the sinus node and the autonomic reflexes that are generated by heart–brain interactions [54]. The RR interval itself is clinically used to calculate HRV with pure heartbeats. The time flux between heartbeats may be inconsistent, varying within a range of 10–30%, with an inconsistency known as HRV [54]. In PD studies, RR interval and HRV analysis are conducted to investigate autonomic modulation of cardiac activity in the patients. Therefore, dysautonomia such as orthostatic hypotension has been associated with these parameters [45,46,47]. Due to its convenience of application to patients, HRV is an established index in PD research [44,45,47,48,49]. There are a wide range of approaches to interpreting HRV parameters associated with PD [44,45]. For example, Alonso et al. (2018) estimated the hazard ratio (HR) using the values of the standard deviation of the normal-to-normal (SDNN) RR interval and root mean square of successive differences (rMSSD) in normal-to-normal RR intervals. In their cohort-group, decreased SDNN and rMSSD were correlated with an increased risk of PD [45]. Power spectral analysis of HRV can reflect parasympathetic and sympathetic activities in PD patients. Strano et al. (2015) detected a reduction in total power in the low frequency (LF) and high frequency (HF) component of HRV in PD patients (n = 18) [48]. Gibbons et al. (2017) found that decreased HRV were associated with orthostatic blood pressure in PD patients, even if they did not detect a significant association between HRV and PD severity [50]. Since HRV could be correlated with PD, it could have potency as a biomarker for PD in the future. However, conversely, a recent study by Akbilgic et al. (2020) observed that there were no significant HRV differences between prodromal PD patients and the healthy cohort [25], which might imply limited use due to low sensitivity in specifically prodromal PD patients. Other reports have suggested a correlation between the RR interval and patients’ severity of PD using a diagnostic index, such as the Hoehn–Yahr (H–Y) scale [55]. Shibata et al. (2009) studied the coefficient variance of RR interval (CVR-R) as an indicator for parasympathetic activity. Although CVR-R was not related to the H–Y stage or the age of the patients; instead, CVR-R was significantly related to heart-to-mediastinum (H/M) ratio in the ^123^iodine-metaiodobenzylguanidine (MIBG) scintigraphy test [49].

*PR interval* is calculated as the time of conduction from the sinus node to the ventricle. An abnormal PR interval may be caused by a conduction block in the atrium or the atrioventricular (AV) node. Generally, a mild degree (first-degree) of atrioventricular conduction block is considered benign in clinical practice [56]. On the other hand, some long-term patient observation reports point out that a prolonged PR interval might be associated with increased cardiovascular risks, such as atrial fibrillation (AF) [57,58]. To date, one research group reported that PR intervals were significantly prolonged in PD patients [52]. Additionally, the PR interval negatively correlated with early and delayed H/M ratios in MIBG scintigraphy. The authors concluded that the prolongation of the PR interval may be caused by a reduction of cardiac sympathetic activity or sympathetic degeneration. However, there were no significant alterations in other ECG parameters in their study. Moreover, considering that other research groups have failed to identify any significant changes on the PR interval, the alteration of the PR interval may originate from the different composition of the patient cohort groups [28,53].

*QT interval* recapitulates the period from emergence of depolarization to termination of repolarization [59]. Alteration in the corrected QT interval (QTc) is associated with dysautonomia [60]. To date, the importance of the QT interval has been reported in several PD patient case studies [28,49,50,51,53]. In these reports, the interrelationship of the QT interval and the disease severity of PD patients were investigated. Gibbons et al. (2017) found that a prolonged QT interval was correlated with earlier onset PD (aged < 57 years) and greater PD progression (*p* < 0.05) [50]. Deguchi et al. (2002) reported that QTc was significantly prolonged in PD patients compared to healthy controls (*p* < 0.001), while the RR and QT intervals were not correlated with PD [46]. Gibbons et al. (2017), Zhong et al. (2021), and Piqueras-Flores et al. (2009) all observed that a prolonged QT interval is positively correlated with the H–Y scale, supporting the notion that QT interval prolongation is associated with the disease severity of PD patients [28,50,53]. Mochizuki et al. (2017) reported the significant correlation between the QRS interval and H–Y scale in their 160 PD patient cohort, with or without anti-Parkinsonian drugs, albeit they failed to observe a significance between the QT interval and H–Y scale [51].

Interestingly, alterations on the ECG in case studies did not display the same pattern among the reports. For instance, while Piqueras-Flores et al. (2018) did not find a significant alteration on the PR interval, Mochizuki et al. (2015) reported prolonged a PR interval in PD patients. The disparate findings on ECG interval alterations may be due to the heterogeneity of the patient cohort group, including the differences in anti-Parkinsonian therapy. Contrary to the study by Mochizuki et al., Zhong et al. surveyed patients that had stopped anti-Parkinsonian therapy. In contrast, Piqueras-Flores investigated PD patients on anti-Parkinsonian therapy, but concluded that prolongation of the QT interval occurred in PD patients regardless of their medication history (including domperidone, antidepressants or antipsychotic agents). Ruonala et al. (2015) reported that the anti-Parkinsonian drug (L-dopa) caused an effect on patients’ ECG pattern during the L-dopa challenge test [61]. However, another study by Cunnington et at. (2013) reported a relationship between the drugs and increased QT interval. It is thus still controversial whether the prolonged QT interval in PD patients is dependent or not upon medication [62]. In addition, Piqueras-Flores et al. found that the left ventricular mass index (*p* = 0.003) and left atrial volume (*p* = 0.01) were also increased with a longer QT interval, which could explain the high prevalence of HF in PD patients. Conclusively, it has been established that PD does affect the patient’s ECG patterns; however, it may not be easy to recognize any specific patterns in global PD patients due to the different nature of cohorts in individual studies. The irregular heartbeat might stem from the alteration of autonomic nervous tone or poor conduction within the heart (channelopathy), thereby emphasizing the need for more fundamental studies that can elaborate on these important clinical findings.

### 4.4. Sudden Unexpected Death in Parkinson’s Disease (SUDPAR)

Indisputably, alterations of cardiac physiology may have fatal consequences. Particularly, sudden unexpected death is overrepresented in PD patients. Therefore, cardiac risks in PD should not be underestimated. In most cases of sudden unexpected death, the leading causes are within the cardiovascular system, respiratory system, and central nervous system (CNS) [30]. Unfortunately, in spite of advances in Parkinsonism drugs, there is still an increased rate of premature death in PD patients compared to the general population, and a considerable number of SUDPAR cases have been documented in case studies [63]. Moreover, autopsy reports indicated that from 5% to 43.7% of PD patients had died suddenly and unexpectedly [30,31,33]. The pathogenesis of SUPDAR is currently unexplained; however, both cardiac abnormalities and autonomic dysfunction may directly contribute to SUDPAR as reviewed by Nejm et al. (2019) and Scorza et al. (2022) [30,64]. On the contrary, other research groups claim that dysautonomia might not be a primary reason for sudden cardiac death in PD [32,65]. Nishida et al. (2017) reported that patients dying with SUPDAR presented with both cardiomyopathy and vasculopathy [32]. The cause of sudden death in PD patients is still unclear; however, multiple studies suggest that cardiovascular risk may play an important role [31,32,63].

## 5. Alpha-Synuclein and Neurocardiac Features in Parkinson’s Disease

### 5.1. Pathological Findings in the Heart: Is It Neurogenic or Intrinsic to the Cardiac Nervous System?

Post-mortem studies have yielded insights into the pathogenesis and have raised a fundamental question for PD research: whether the cardiac-associated abnormalities in PD are propagated from the autonomic nervous system or are intrinsic to the cardiac nervous system. This question is still open. The pathological findings observed in PD were illustrated in Figure 2. Nishida et al. (2017) investigated two autopsy cases of SUDPAR, and observed ventricular wall thickness, septal fibrosis, and severe stenosis of the atrioventricular node artery, which might be a suggestive sign of autonomic dysregulation [32]. However, it is unclear whether the cardiac autonomic nervous system (ANS) and cardiomyopathy contribute more to PD-associated features.

### 5.2. Cardiac Dysautonomia in the Heart of Parkinson’s Disease Patients

Dysregulation of cardiac electrophysiology can be caused by cardiac autonomic innervation or cardiomyopathy [22]. Autonomic dysregulation associated with PD has been suggested to be ubiquitous [22], and a body of literature indicates that Lewy bodies, a hallmark of PD pathology, may play a role in cardiac dysautonomia [23]. For example, OH (orthostatic hypotension) is the most common cardiac dysautonomia symptom in PD, affecting about 50–60% of PD patients [23,39]. OH is associated with Lewy body pathology in the peripheral nervous system [23,66]. Interestingly, previous studies unanimously indicated that multiple mechanisms could contribute to cardiac dysautonomia in PD patients. First, cardiac autonomic denervation followed by PD-mediated neurodegeneration; second, loss of extra-cardiac sympathetic adrenergic innervation; third, decreased parasympathetic parameters; and fourth, reduced sympathetic and parasympathetic functions of arterial baroreflex [34,67,68]. Cardiac dysautonomia in PD patients has been pathologically interrogated by several research groups [69,70]. Amino et al. (2005) conducted a quantitative analysis to try to determine and map the degeneration of nerves within PD patients’ hearts. They compared the reduction of tyrosine hydroxylase positive areas in the cardiac tissue to that of overall nerve degeneration in the cardiac tissue and found that the degeneration was more prominent in the cardiac sympathetic neurons in PD patients [69]. Takahashi et al. (2016) elaborated that cardiac dysautonomia can be correlated with the delay of ^123^I-MIBG uptake in PD patients and concluded that cardiac denervation in PD occurred via the sympathetic nervous system [70]. However, this second study did not see a selective sympathetic nervous degeneration as seen in Amino et al., rather it found that cardiac neurodegeneration occurred not only in sympathetic nerves but also in most of the neurons in the study; Takahashi’s conclusion might be premature. It is still controversial whether cardiac dysautonomia originated from peripheral sympathetic loss. Nonetheless, the evidence raises the possibility that cardiac dysautonomia in PD can be directly associated with pathological changes of cardiac innervation from the autonomic nervous system.

### 5.3. Cardiac Nerves as a Potential Channel for the Propagation of Lewy Body

Braak (2003) hypothesized that Lewy bodies might propagate from dorsal motor nuclei to the whole limbic system [3]. Since then, significant advances have been made to support his idea. In the last decades, NMS research has focused on the neuronal pathologies developing in autonomic peripheral ganglions in intestine and skin, which are physically linked to dorsal nuclei via vagal nerves [71,72].

The Lewy body-mediated neurodegeneration in PD patients can be assigned to the aggregation of α-synuclein in the neuronal cells. Interestingly, α-synuclein mediated pathology (α-synucleinopathy) has been known to be transmittable via the nervous system [73]. Since the ANS is a major regulator of neurocardiac physiology, cardiac lesions found in the PD heart can be critical for interrogating the direction of neurodegenerative propagation that PD patients may undergo. Studies have revealed that α-synucleinopathy can develop in cardiac intrinsic nerves [32,35,36]. Nishida et al. (2017) confirmed that SUDPAR patients had an α-synucleinopathy in the heart, consistent with the brain pathology. Interestingly, Navarro-Otano et al. (2013) observed that α-synucleinopathy can develop in epicardial fat tissues from patients without any Parkinsonism, but with a significantly correlated prevalence of NMS such as constipation or sleep disorder [35]. Moreover, it was observed that some of the tissues were positive to the sympathetic marker tyrosine hydroxylase, implying that sympathetic dysregulation could develop with the occurrence of α-synucleinopathy [35]. In addition, Javanshiri et al. (2022) investigated a large number of autopsy cases diagnosed with Lewy body diseases (PD, multisystem atrophy, and Lewy body dementia); the prevalence of cardiac α-synucleinopathy was significantly higher in the Lewy body dementia presenting group (*p* < 0.001) compared to a healthy control group. According to the report, 82% of Lewy body presenting patients (56 of 68 subjects) showed cardiac α-synucleinopathy, while none of the 32 healthy controls did [36]. Remarkably, α-synucleinopathy was unevenly located in the heart; not all the nerve cells were positive for α-synuclein in the tissue sample. Interestingly, Javanshiri et al. (2013) further investigated that the presence of cardiac α-synucleinopathy is positively correlated with the prevalence of sudden cardiac death [74]. The evidence indicates the PD patients may undergo neurodegeneration in the heart which was mediated by α-synuclein.

In conclusion, cardiac pathology in PD may hold critical information with respect to the propagation of neurodegeneration. Recently, it has been revealed that both sympathetic and parasympathetic nerves are susceptible to α-synucleinopathy, causing autonomic dysregulation in mice [75]. However, the direction and the major propagating nerve types are still unclear. The direction could be centripetal, centrifugal, or bidirectional, and the dominantly affected nerves could be sympathetic or parasympathetic. Seemingly, cardiac pathologies can play a critical role in deciding these directions.

## 6. Electrophysiology in Parkinson’s Disease Models

### 6.1. Electrophysiological Mechanism of Observed Cardiac Related Changes

As discussed earlier in this review, it is evident that PD can cause alterations of the cardiac electrophysiological phenotype in patients. However, the lack of details elucidating the underlying mechanism of how PD affects the heart remains a barrier to understanding cardiac pathologies in PD.

### 6.2. Animal Models of Parkinson’s Disease and Their Manifestation of Cardiac Electrophysiology Abnormalities

In humans, mutations of specific genes associated with Parkinsonism or Lewy body pathology can affect cardiac electrophysiology indicators such as HRV [76,77]. Balestrini et al. (2020) reported the correlation between rapid-onset dystonia-Parkinsonism (RDP) and prevalence of ECG abnormalities in patients with RDP [76]. They generated an in vivo model using a knock-out of ATP1A3 and confirmed that ECG abnormalities and the exacerbated seizure-induced cardiac arrest were replicated in the transgenic mice. In addition, mice overexpressing α-synuclein exhibited HRV impairment and abnormal cardiac rhythm, accompanied by a reduction of midbrain dopaminergic cells [77]. The result may be translated as a consequence of PD. However, so far, no research has been published demonstrating cardiac electrophysiology alteration in disease models with established PD risk factor genes [78].

### 6.3. Patch Clamp Electrophysiology in In Vitro Models of Parkinson’s Disease

Patch clamp electrophysiology is a powerful tool enabling measurement of ion currents in cardiomyocytes [79]. To date, there is a lack of suitable in vitro models for cardiac electrophysiology; however, there are available studies using in vitro models of murine primary microglial cells, human pluripotent stem cell (iPSC)-derived midbrain dopaminergic cells, dendritic cells, the human embryonic kidney 293 (HEK293) cell line, and the PC12 cell line, indicating that PD-associated proteins could change electrophysiological properties in vitro (Figure 3) [80,81,82,83].

Whole-cell patch clamp recordings demonstrated that stimulation with the direct application of the aggregated α-Synuclein protein caused upregulation of voltage-gated potassium channel Kv1.3 expression and activity of the outward Kv1.3 current in primary cultured mouse microglial cells [80] (Figure 3A). The authors confirmed the functional relevance of α-synuclein-medicated neuroinflammation and the Kv1.3 channel by using the Kv1.3 knock-out cells and using the Kv1.3 inhibitor in vitro. In addition, Virdi et al. (2022) investigated human-induced pluripotent stem cells (iPSCs) originating from PD patients having the α-synuclein A53T point mutation or chromosomal triplicate. The mutation of α-synuclein affects resting membrane potential in midbrain dopaminergic neuronal cells, which results in impaired firing of the neuron [83].

These findings of changes in microglia properties raise the possibility that neuroinflammation may change the electrophysiological properties in different cell types by interrupting ion channel regulation. Moreover, Bedford et al. (2016) investigated that leucine-rich repeat kinase 2 (LRRK2), a well-known familial PD risk factor gene, could increase the activity of the inward Cav2.1 current in HEK293 cells [81] (Figure 3B). The HEK293 cells were transfected with Cav2.1 alone or together with LRRK2. The inward calcium currents were increased via direct protein interactions between the Cav2.1 channel and LRRK2 protein [81]. They also examined a neuroendocrine rat pheochromocytoma PC12 cell line transfected with LRRK2; the activity of endogenous calcium channels was increased when transfected with LRRK2 [81]. Hosseinzadeh et al. (2016) examined primary cultured dendritic cells from the LRRK2 knock-out (*Lrrk2*^−/−^) and overexpressing (*Lrrk2*^+/+^) mice [82], and found that the Na^+^/K^+^-ATPase current was increased in LRRK2 overexpressing cells with an increase of the channel expression on protein level when compared to that of knock-out cells. To sum up, there are indications through electrophysiology studies that suggest a significant effect of PD-associated pathologies in in vitro models, but further studies are needed to link this definitively to cardiac clinical pathology.

## 7. Conclusions

Cardiovascular abnormalities in PD patients have increasingly been recognized. Recent studies have revealed the involvement of various cardiovascular diseases in PD. In this review, we have summarized the increased cardiovascular risks, cardiac arrhythmic conditions, cardiomyopathies, and neurocardiac lesions found in PD patients. Furthermore, we have highlighted the data reporting the presence of α-synuclein accumulation in cardiac tissue and patients’ alterations in neurocardiac regulation. Based on the cardiovascular-related features of PD, we propose that there may be substantial changes in the electrophysiological activity and remodeling of the myocardium, in the hearts of PD patients [22,78]. However, a complete electrophysiological understanding of the heart, and the cardiac autonomic nervous system, remain to be determined due to a lack of translational studies. Although researchers have conducted animal experiments by surgically or chemically inducing PD symptoms, the studies might not fully recapitulate the nature of actual patients, considering that non-motor symptoms develop in the prodromic or early phase of PD. Here, we have reviewed the existing in vitro models that utilize α-synuclein protein aggregation or gene mutation to investigate the impact of pathology on cardiac tissue electrophysiology. However, we found a lack of published in vitro models addressing α-synuclein-mediated Parkinson’s disease in cardiac cells. Closing this research gap between clinical observations and our understanding of PD-related cardiac remodeling is crucial. Furthermore, given the significance of the disease, further research is needed to explore the relationship between cardiac electrophysiology and Parkinson’s disease.

## Figures and Tables

**Figure 1 ijms-24-12601-f001:**
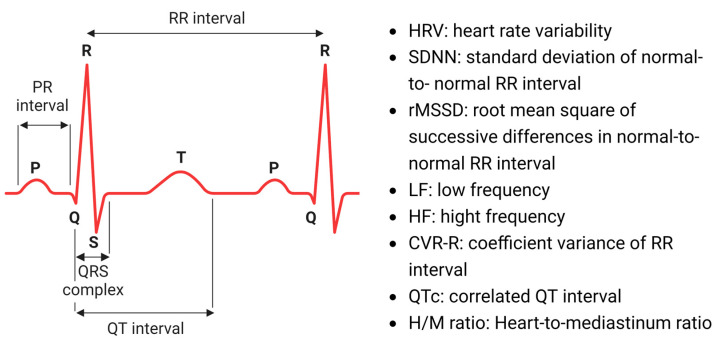
Illustration of electrocardiographic signals and parameters. Images were created with BioRender.

**Figure 2 ijms-24-12601-f002:**
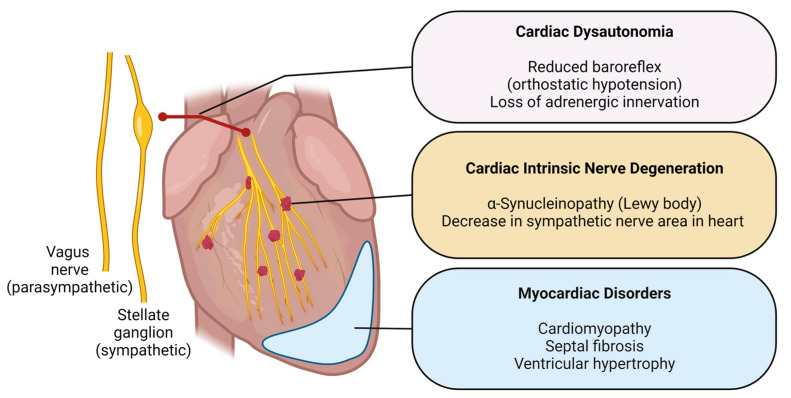
Illustration of PD-associated cardiac pathologies. Images were created with BioRender.

**Figure 3 ijms-24-12601-f003:**
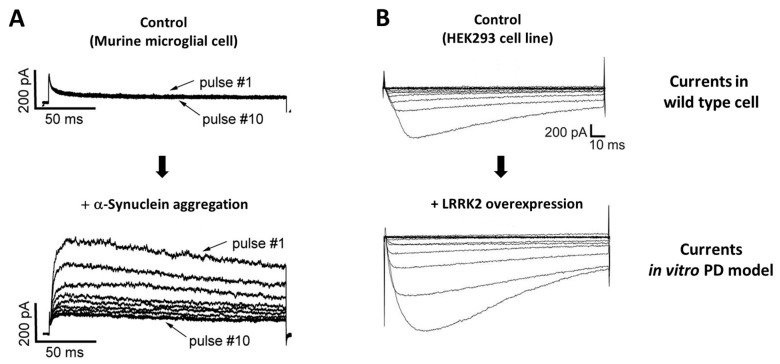
Parkinson’s disease-related changes in cell currents in vitro disease models. Whole-cell patch clamp recording of outward voltage-gated potassium channel Kv1.3 current in murine primary microglial cells treated with α-Synuclein aggregation for 48 h (Sarkar et al., 2020 [80]) (**A**). α-synuclein aggregation-treated cells revealed increased Kv1.3 activity with microglial inflammation. Each current was measured by applying 200-ms pulses from −80 to 40 mV. The selectivity of the Kv1.3 current was pharmacologically verified. Whole-cell inward voltage-gated calcium channel Cav2.1 current recorded from in vitro Parkinson’s disease model with human embryonic kidney 293 (HEK293) cell (Bedford et al., 2016 [81]) (**B**). The leucine rich repeat kinase 2 overexpression increased the activity of the Cav2.1 channel in HEK293 cells. The current was recorded with a voltage-step increasing protocol. The Cav2.1 was transfected with green fluorescent protein.

**Table 1 ijms-24-12601-t001:** Cardiovascular-Associated Pathologies in Parkinson’s Disease.

Category	Symptoms	References
Cardiovascular Risks of PD Patients	Stroke	[22,24]
	Coronary artery disease	[22,24]
	Acute myocardial infarction	[25,26]
	Increase in Cardiovascular Risk Score (QRISK)	[27]
	Hypolipidemia	[28]
	Heart failure	[29]
	Sudden cardiac death (SCD)	[30,31]
2. Electrocardiographic (ECG) alterations in PD Patients	Heart rate alterations	Summarized in Section 4.3
	Arrythmias - RR interval alteration - PR interval alteration - QT interval alteration	
3. Neurocardiac and myocardial lesions	Cardiomyopathy	[28,32]
	Septal fibrosis	[32,33]
	Stenosis in atrioventricular node	[32]
	Ventricular hypertrophy	[28,32]
	Decrease in cardiac sympathetic nerve area	[34]
	Cardiac α-synucleinopathy (Lewy Body)	[35,36]

**Table 2 ijms-24-12601-t002:** Electrocardiographic alterations observed in Parkinson’s disease patients.

Parameters (ECG)	Author (Year)	Indicator	Cohorts/Controls	Reference
RR interval / Heart Rate Variability (HRV)	Heimrich et al. (2021)	Decrease in rMSSD (*p* < 0.01)	1566 PD patients/1206 controls (meta-analysis)	[44]
	Alonso et al. (2015)	rMSSD (HR 95% CI 2.1), SDNN (HR 95%, CI 2.9)	78 incidental developed PD during follow-up/12,162 community-based cohort without PD	[45]
	Deguchi et al. (2002)	RR and heartrate were not correlated with PD	34 (17 male/17 female, mean duration 5.8 years) PD patients/30 controls	[46]
	Haensch et al. (2007)	HRV was not correlated with PD	58 (27 male, 31 female) PD patients (mean duration 5.1 years, drug naïve)	[47]
	Strano et al. (2016)	Power spectral analysis of HRV (reduced in PD patients, *p* < 0.001)	18 newly diagnosed PD patients, (drug naïve)/8 patient without PD	[48]
	Shibata et al. (2009)	CVR-R (correlated with MIBG uptake; early stage, r = 0.457, *p* < 0.001; late stage, r = 0.442, *p* < 0.001)	79 (39 male, 40 female) PD patients	[49]
	Gibbons et al. (2017)	HRV was not correlated with severity or duration of PD. Baseline HRV was lower in the old group than age younger than 57	1741 PD patients/Patient’s baseline ECG before 5 years follow-up	[50]
	Mochizuki et al. (2017)	RR was not correlated with PD	156 (66 male, 90 female) PD patients (mean duration 1.9 years, drug naïve)	[51]
PR interval	Piqueras-Flores et al. (2017)	PR interval was not correlated with PD	50 idiopathic PD patients/50 control	[28]
	Mochizuki et al. (2017)	PR was not correlated with PD	Mentioned above	[51]
	Mochizuki et al. (2015)	Prolonged PR interval (PD vs. normal group, *p* < 0.05)	191 PD patients (drug naïve)/124 controls	[52]
	Zhong et al. (2021)	PR interval was associated with PD severity (HY stage, r = 0.72, *p* < 0.01) and disease progression (r = 0.72, *p* < 0.001)	118 PD patients (74 postural instability and gait difficult PD/44 tremor-dominant PD)	[53]
QT interval and QRS complex	Deguchi et al. (2002)	QTc was prolonged (*p* < 0.001) in PD patients, QT intervals was not correlated with PD	Mentioned above	[46]
	Shibata et al. (2009)	QT interval was not correlated with PD		[49]
	Gibbons et al. (2017)	QR interval was correlated with PD severity (GST score, *p* < 0.05)		[50]
	Mochizuki et al. (2017)	QRS are correlated with PD severity (HY stage, r = 0.206, *p* < 0.01)		[51]
	Zhong et al. (2021)	QRS, QTc, and QT interval was positively correlated with PD severity (HY stage, QRS: r = 0.82, *p* < 0.01; QTc: r = 0.21, *p* < 0.05′ QT interval: r = 0.24, *p* < 0.01) and disease progression (QRS: r = 0.43, *p* < 0.01; QTc: r = 0.76, *p* < 0.001; QT interval: r = 0.55, *p* < 0.01)		[53]

RR, R to R interval; HRV, heart rate variability; SDNN, standard deviation of normal to normal RR interval; rMSSD, root mean square of successive differences in normal to normal RR interval; CVR-R, coefficient variance of RR interval; QT, Q to T interval; QTc, correlated QT interval; GST, global statistical test; HY, Hoehn and Yahr scale; *p*, *p*-value; r, correlation coefficient.

## Data Availability

Data management was managed followed by “MDPI Research Data Policies” at https://www.mdpi.com/ethics (accessed on 25 May 2023).

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
