# Peer review of "Clinical and Non-Clinical Cardiovascular Disease Associated Pathologies in Parkinson’s Disease"

_ijms, 2023, doi:10.3390/ijms241612601_

Round 1
Reviewer 1 Report
The authors performed a narrative review on cardiovascular nonmotor symptoms in Parkinson’s disease patients, focusing on cardiovascular disease risks, electrocardiograms (ECG), neurocardiac lesions in PD, and fundamental electrophysiological studies. By reviewing these papers, they have highlighted the involvement of heart in PD pathology, with the heart possibly involved in alfa-synuclein neuronal accumulation and patients undergoing alterations in the neurocardiac regulation.
While the topic is interesting from a research and clinical point of view, also because not well elucidated so far, there are many aspects rendering the review poorly attractive. In particular, the the aim of the study isn’t reported and is not entirely clear, which makes the paper difficult to follow and read. Methodology for adequate article retrieval is lacking and the entire text appears to be a compilation of disjointed excerpts from various reviewed papers, lacking a cohesive narrative and hardly vehiculating a message.
Below some comments regarding some mistakes and unclear sentences:
- “The motor related symptoms may be provoked or aggravated by dopaminergic treatment and the discovery of levodopa as a drug dramatically changed the understanding of PD”.. this sentence needs rephrasing. Motor complications/fluctuations are provoked by dopaminergic treatment (which by the way would still be partly incorrect considering that motor fluctuations do not uniquely depend from dopaminergic treatment), while cardinal motor symptoms (tremor, bradykinesia, and rigidity) are ameliorated by dopaminergic therapy.
- “despite the high prevalence of NMS, the clinical phenotype of NMS varies depending on the individual” and “Hyposmia, impaired vision, hallucinations, pain, anxiety, excessive sweating, depression, cognitive dysfunction, dementia, sleep disturbance, bladder hyperreflexia, and constipation are considered NMS phenotypes in PD.” What do the authors mean by phenotype of nonmotor symptoms? These are common non motor symptoms of PD
- “NMS might be underestimated as less significant complaints because they are relatively negligible compared to the motor symptoms”. Who are they “relatively negligible” for? For the patient? For physicians? Researchers?
- “Conclusively, case-control studies suggested that cardiovascular symptoms may be involved in other NMS via autonomic dysregulation” I don’t understand this sentence.
- “Direct cardiac damage could also be considered as an NMS in PD” and “dysautonomia symptoms such as cardiac reflex and orthostatic hypotension have been associated with these parameters”. Cardiac damage and cardiac reflex or orthostatic hypotension are not symptoms.
- “Since the recognition of cardiac associated consequences of Parkinson’s disease, several studies have aimed to elucidate the cardiac impairment in PD.” What are the cardiac associated consequences of Parkinson’s disease? The sentence isn’t clear.
- “However, current cardiac research also highlights the neuronal systems in the heart, rather than the heart itself, in accordance with the current research trend in NMS”. This sentence is not clear and should be rephrased.
- “Even if the transgenic mice were originally developed to reproduce multiple system atrophy (MSA) symptoms, the result may be translated as a PD consequence due to its motor-dysfunction and a-synucleinopathy.” Again, I do not understand the sentence
-
Minor comments:
- Certain sentences should be rephrased, for example:
o Line 46: “Sporadic PD is typically more late-onset..”
o Line 57: “Therefore, clinical phenotypes of PD are defined by the motor dysfunctions”
o Line 62-63: “Recent studies though have suggested that PD is a neurodegenerative disorder that manifests a constellation of multiple neurological symptoms”
o Line 84: “In most cases of PD, the patients would initially be diagnosed by the motor-related symptoms”
o Line 128: “several studies reported that cardiovascular disease may be highly associated with other NMS and it can be secondary to other NMS symptoms”
o Line 234: “Alteration in correlated QT interval (QTc)”. I think the authors meant the “corrected QT interval”
Some sentences should be rephrased for clarity as reported above.
Author Response
|
General comments |
Revision |
|
The authors performed a narrative review on cardiovascular nonmotor symptoms in Parkinson’s disease patients, focusing on cardiovascular disease risks, electrocardiograms (ECG), neurocardiac lesions in PD, and fundamental electrophysiological studies. By reviewing these papers, they have highlighted the involvement of heart in PD pathology, with the heart possibly involved in alfa-synuclein neuronal accumulation and patients undergoing alterations in the neurocardiac regulation. While the topic is interesting from a research and clinical point of view, also because not well elucidated so far, there are many aspects rendering the review poorly attractive.
|
Thank you for your thorough and helpful suggestions on our manuscript. We appreciate that you recognise the exciting topics we meant to address. As you recommended, coherent-wise, the manuscript can be improved to make it easier to read. We have revised the manuscript on several points with this in mind. First, to improve the coherency of the manuscript, we have added research aim and discussion chapters to guide and recap the topic. Redundant sentences have been removed. Secondly, new references have been added. And thirdly, we have, edited major and minor unclear paragraphs throughout the manuscript. With this revision we believe we have substantially improved the clarity and aim of this review to give full focus on the interesting and important topic.
|
|
In particular, the aim of the study isn’t reported and is not entirely clear, which makes the paper difficult to follow and read. |
Thank you for sharing this concern, we agree and have addressed the issue by adding a separate Aims chapter in the beginning of the review to highlight what the review will cover. “ 1.1. Research Aim
|
|
Methodology for adequate article retrieval is lacking and the entire text appears to be a compilation of disjointed excerpts from various reviewed papers, lacking a cohesive narrative and hardly vehiculating a message.
|
Thank you for your thoughts to add a methodology chapter, however since this is a narrative review we feel that the paper inclusion criteria is flexible and aimed at cover the topic as described in the aims. We would not like to give the impression that this is a systematic review and hence we don’t think it is a good idea to add a methodology chapter. As a whole we hope that the revision with the extensive changes has efficiently addressed the noted lack of cohesiveness. |
|
Major comments |
|
|
this sentence needs rephrasing. “The motor related symptoms may be provoked or aggravated by dopaminergic treatment and the discovery of levodopa as a drug dramatically changed the understanding of PD”.. |
Thank you for this comment we have changed the sentence as below and added a new reference:
New reference:
|
|
“Despite the high prevalence of NMS, the clinical phenotype of NMS varies depending on the individual” and “Hyposmia, impaired vision, hallucinations, pain, anxiety, excessive sweating, depression, cognitive dysfunction, dementia, sleep disturbance, bladder hyperreflexia, and constipation are considered NMS phenotypes in PD.”
|
"Despite the high prevalence of NMS, the manifestation of the symptoms can vary widely across individuals, leading to different clinical presentations. Hyposmia, impaired vision, hallucinations, pain, anxiety, excessive sweating, depression, cognitive dysfunction, dementia, sleep disturbance, bladder hyperreflexia, and constipation are all considered non-motor symptoms commonly associated with PD.” New reference:
|
|
“NMS might be underestimated as less significant complaints because they are relatively negligible |
What we meant was that NMS are relatively difficult to recognise by patients, while motor dysfunction are more easily recognisable. We revised the sentence as below. “Even though NMS are generally highly prevalent among PD patients, the symptoms are often not obvious to the patients themselves as a concomitant problem of their PD [17,18].” (Line 82) New reference Shulman, L.M.; Taback, R.L.; Rabinstein, A.A.; Weiner, W.J. Non-recognition of depression and other non-motor symptoms in Parkinson's disease. Parkinsonism Relat Disord 2002, 8, 193-197, doi:10.1016/s1353-8020(01)00015-3. |
|
“Conclusively, case-control studies suggested that cardiovascular symptoms may be involved in other NMS via autonomic dysregulation” |
We have rewritten the sentence to make it clearer what we mean. “Cardiovascular symptoms can be a direct effect of PD, in addition, case-control studies suggest that it can also be a side effect of autonomic dysfunction (dysautonomia)-associated NMS. In this regard, understanding cardiovascular association with PD may help shed light on the systemic autonomic dysregulation of PD in general. (Line 142)
|
|
“Direct cardiac damage could also be considered as an NMS in PD” and “dysautonomia symptoms such as cardiac reflex and orthostatic hypotension have been associated with these parameters”.
|
Admittedly, cardiac damage and cardiac reflex are not well elucidated in PD, therefore cardiac associated features would be a word to explain these (Schapira et al, 2017, Non-motor features of Parkinson Disease, Nat Rev Neurosci, doi.org/10.1038/nrn.2017.62). In addition, these cardiac disease are perhaps more a consequence of the disease, or a separate diagnosis. We have revised the sentence as below to avoid confusion and clarify the cardiac features identified in PD “Parkinson’s disease patients develop cardiac abnormalities such as cardiomyopathies or heart failure.” (Line 146) However, when explaining orthostatic hypotension, it has been described as a non-motor symptoms in PD (Chaudhuri et al., 2006, Non-motor symptoms of Parkinson’s diseases: diagnosis and management, Lancet Neurology,doi.org/10.1016/S1474-4422(06)70373-8). Thus, we described orthostatic hypotension as a non-motor symptom in this manuscript.
|
|
“Since the recognition of cardiac associated consequences of Parkinson’s disease, several studies have aimed to elucidate the cardiac impairment in PD.” |
We apologise for being unclear. The cardiac associated consequences refer to what was discussed in chapter 4.1: cardiac functional change (ECG) as a concomitant consequence of cardiac abnormalities following PD. We have edited the sentence to clarify: “Since the recognition of cardiac associated consequences such as electrophysiological alterations following PD development, several studies have aimed to further elucidate cardiac impairments caused by PD.” (Line 173)
|
|
“However, current cardiac research also highlights the neuronal systems in the heart, rather than the heart itself, in accordance with the current research trend in NMS”. |
Rephrased to: “However, it is unclear whether the cardiac autonomic nervous system (ANS) and cardiomyopathy contribute more to PD-associated features.” (Line 297)
|
|
“Even if the transgenic mice were originally developed to reproduce multiple system atrophy (MSA) symptoms, the result may be translated as a PD consequence due to its motor-dysfunction and a-synucleinopathy.”
|
We apologise for being unclear. The researcher meant to design the study for investigating MSA, however, the mechanism of disease model is the same with that of Parkinson’s disease animal model using a-synuclein overexpressing. Explaining MSA model might not be necessary here. “In addition, mice overexpressing a-synuclein exhibited HRV impairment and abnormal cardiac rhythm, accompanied by a reduction of midbrain dopaminergic cells [74]. The result may be translated as a consequence of PD. (Line 380)
|
|
Minor comments |
|
|
Line 46: “Sporadic PD is typically more late-onset..” |
“Sporadic PD is typically more late-onset than familial PD and is considered the most frequent type of the disease, while familial PD may develop earlier and is associated with mutation of risk factor genes” (Line 54) |
|
Line 57: “Therefore, clinical phenotypes of PD are defined by the motor dysfunctions” |
“Clinically patients are diagnosed by the motor dysfunctions [8].” (Line 64) |
|
Line 62-63: “Recent studies though have suggested that PD is a neurodegenerative disorder that manifests a constellation of multiple neurological symptoms” |
“Recent studies though have indicated that PD is a constellation of multiple neurological symptoms beyond the classic triad of Parkinsonism [10].” (Line 68) |
|
Line 84: “In most cases of PD, the patients would initially be diagnosed by the motor-related symptoms” |
“In most cases of PD, the patients are diagnosed with PD when they develop motor symptoms, however, these motor symptoms are generally only recognised when patients have already lost 60% - 80% of the midbrain dopaminergic neurons [21,22].” (Line 90) |
|
Line 128: “several studies reported that cardiovascular disease may be highly associated with other NMS and it can be secondary to other NMS symptoms” |
“Several studies have reported that cardiovascular disease may be associated with and present together with other NMS and it can also be present as a consequence of other NMS. [38,39]. (Line 133)
|
|
Line 234: “Alteration in correlated QT interval (QTc)”. I think the authors meant the “corrected QT interval”
|
It should be “corrected QT.” (Line 239) Thank you.
|

Reviewer 2 Report
The review should contain more references.
Please add a chapter with discussions, trying to conclude all the information presented.
Also add a chapter after the introduction presenting the Material and Methods, also adding inclusion and exclusion criteria etc.
Author Response
|
Comments |
|
|
Comments and Suggestions for Authors The review should contain more references.
|
We appreciate and thank you for your suggestions to improve our manuscript. We have revised accordingly; added Discussion chapter and more references. In addition a more clear research aim and other clarifications in the text has been added to make the manuscript more coherent and readable. The revised sentences are marked in red.
New References: Thanvi, B.R.; Lo, T.C. Long term motor complications of levodopa: clinical features, mechanisms, and management strategies. Postgrad Med J 2004, 80, 452-458, doi:10.1136/pgmj.2003.013912.
Chaudhuri, K.R.; Martinez-Martin, P.; Brown, R.G.; Sethi, K.; Stocchi, F.; Odin, P.; Ondo, W.; Abe, K.; Macphee, G.; Macmahon, D.; et al. The metric properties of a novel non-motor symptoms scale for Parkinson’s disease: Results from an international pilot study. Mov Disord 2007, 22, 1901-1911, doi:10.1002/mds.21596.
Global Parkinson's Disease Survey Steering, C. Factors impacting on quality of life in Parkinson's disease: results from an international survey. Mov Disord 2002, 17, 60-67, doi:10.1002/mds.10010.
Nejm, M.B.; Andersen, M.L.; Tufik, S.; Finsterer, J.; Scorza, F.A. Sudden death in Parkinson's disease: Unjustifiably forgotten. Parkinsonism Relat Disord 2019, 58, 88-89, doi:10.1016/j.parkreldis.2018.08.012.
Zhang, Y.; Wang, G. Response to: Sudden death in Parkinson's disease: Unjustifiably forgotten. Parkinsonism Relat Disord 2019, 58, 87, doi:10.1016/j.parkreldis.2018.08.013 Javanshiri, K.; Drakenberg, T.; Haglund, M.; Englund, E. Sudden cardiac death in synucleinopathies. J Neuropath Exp Neur 2023, 82, 242-249, doi:10.1093/jnen/nlad001. Shulman, L.M.; Taback, R.L.; Rabinstein, A.A.; Weiner, W.J. Non-recognition of depression and other non-motor symptoms in Parkinson's disease. Parkinsonism Relat Disord 2002, 8, 193-197, doi:10.1016/s1353-8020(01)00015-3.
|
|
Please add a chapter with discussions, trying to conclude all the information presented.
|
Thank you for the suggestion, we have added a concise discussion that summarises the presented data in the review, pinpoints some of the current challenges, and gives our conclusions on the topic.
7. Discussion “Cardiovascular risks have been recognized as non-motor symptoms in PD, and cardiovascular risks, such as hypertension or postural hypotension, are considered as a diagnostic parameter of NMS [14]. Recent year studies have revealed the involvement of various cardiovascular diseases in PD. In this review we have summarised the increased cardiovascular risks, cardiac arrhythmic conditions, cardiomyopathies, and neurocardiac lesions found in PD patients. Further we have highlighted the presence of a-synuclein accumulation in the cardiac tissue and patients undergoing alterations in the neurocardiac regulation. Based on the cardiovascular-related features of PD, we conclude to anticipate substantial changes in the electrophysiological activity and remodelling of the myocardium in the hearts of PD patients. [24,77]. However, complete electrophysiological understanding of the heart and the cardiac autonomic nervous system remain unclear due to lack of translational studies. Although researchers have conducted animal experiments by mechanically or chemically inducing PD symptoms, the studies might not fully recapitulate the nature of actual patients considering that non motor symptoms develop in prodromic or early phase of PD. Here we have reviewed the existing in vitro models that utilize α-synuclein protein aggregation or gene mutation to investigate the impact on tissue electrophysiology. However, we found a lack of published in vitro models addressing the α-synuclein-mediated Parkinson's disease in cardiac cells. Closing this research gap between clinical observations and our understanding of PD-related cardiac remodelling is crucial. Furthermore, given the significance of the disease, further research is needed to explore the relationship between cardiac electrophysiology and Parkinson's disease. (Line 431)
|
|
Also add a chapter after the introduction presenting the Material and Methods, also adding inclusion and exclusion criteria etc.
|
Thank you for your thoughts to add a methodology chapter, however since this is a narrative review, we feel that the paper inclusion criteria is flexible and aimed at cover the topic as described in the aims. We would not like to give the impression that this is a systematic review and hence we don’t think it is a good idea to add a methodology chapter.
|

Reviewer 3 Report
Clinical and Non-Clinical Cardiovascular Disease Associated 2 Pathologies in Parkinson’s Disease
The topic chosen about non-motor pathology in Parkinson's disease is particularly interesting. References must be updated.References must be updated. The tables and images are important and are meticulously made.Author Response
Reviewer 3
|
Comments |
Revision |
|
The topic chosen about non-motor pathology in Parkinson's disease is particularly interesting. References must be updated. The tables and images are important and are meticulously made.
|
Thank you for your suggestions to help improve our manuscript. We have added more references, see list below.
Aside from the references, we have added new chapters, and revised the manuscript to make it clearer and more coherent. The revised sentences are marked in red.
New References: Thanvi, B.R.; Lo, T.C. Long term motor complications of levodopa: clinical features, mechanisms, and management strategies. Postgrad Med J 2004, 80, 452-458, doi:10.1136/pgmj.2003.013912.
Chaudhuri, K.R.; Martinez-Martin, P.; Brown, R.G.; Sethi, K.; Stocchi, F.; Odin, P.; Ondo, W.; Abe, K.; Macphee, G.; Macmahon, D.; et al. The metric properties of a novel non-motor symptoms scale for Parkinson’s disease: Results from an international pilot study. Mov Disord 2007, 22, 1901-1911, doi:10.1002/mds.21596. Global Parkinson's Disease Survey Steering, C. Factors impacting on quality of life in Parkinson's disease: results from an international survey. Mov Disord 2002, 17, 60-67, doi:10.1002/mds.10010. Zhang, Y.; Wang, G. Response to: Sudden death in Parkinson's disease: Unjustifiably forgotten. Parkinsonism Relat Disord 2019, 58, 87, doi:10.1016/j.parkreldis.2018.08.013 Javanshiri, K.; Drakenberg, T.; Haglund, M.; Englund, E. Sudden cardiac death in synucleinopathies. J Neuropath Exp Neur 2023, 82, 242-249, doi:10.1093/jnen/nlad001.
|

Round 2
Reviewer 1 Report
The Authors have addressed many of my previous comments or concerns.
The quality and readability of the paper have improved, and some mistakes have been corrected.
There is still a mistake in the new version of the Discussion:
"Cardiovascular risks have been recognized as non-motor symptoms in PD, and cardiovascular risks, such as hypertension or postural hypotension, are considered as a diagnostic parameter of NMS."
Risks are not symptoms. Moreover, hypertension is not a 'diagnostic parameter' of NMS in PD. Please, correct the entire sentence.
Minor corrections needed
Author Response
We have revised the sentence considering your comment. The new sentence is as below.
“Cardiovascular abnormalities in PD patients have increasingly been recognised. Recent studies have revealed the involvement of various cardiovascular diseases in PD.” (Line 440)